# Shoaled glacial AMOC despite vigorous tidal Dissipation: Vertical Stratification matters

Yugeng Chen[1,2], Pengyang Song[1], Xianyao Chen[2,3], Gerrit Lohmann[1,4]

[1]Alfred Wegener Institute, Helmholtz Center for Polar and Marine Research, Bremerhaven, 27570, Germany
[2]Frontier Science Center for Deep Ocean Multispheres and Earth System / Physical Oceanography Laboratory, Ocean University of China, Qingdao, 266100, China
[3]Laoshan Laboratory, Qingdao, 266100, China
[4]University of Bremen, Bremen, 28334, Germany

*Correspondence to*: Yugeng Chen (yugeng.chen@awi.de)

**Abstract.** During the Last Glacial Maximum (LGM), tidal dissipation was about threefold higher than today, which could have led to a considerable increase in vertical mixing. This would enhance the glacial Atlantic Meridional Overturning Circulation (AMOC), contradicting the shoaled AMOC as indicated by paleo proxies. Here, we conduct ocean model simulations to investigate the impact of background climate conditions and tidal mixing on the AMOC during the LGM.   We successfully reproduce the stratified ocean characteristic of the LGM by accurately simulating the elevated salinity of the deep
sea and the rapid temperature decrease in the ocean's upper layers. Our findings indicate that the shoaled glacial AMOC is mainly due to strong glacial ocean stratification, irrespective of enhanced tidal dissipation. However, glacial tidal dissipation plays a critical role in the intensification of the AABW during the LGM. Given the critical role of the AMOC in (de-)glacial climate evolution, our results highlight the complex interactions of ocean stratification and tidal dissipation that have been neglected so far.

## 1 Introduction

The Atlantic meridional overturning circulation (AMOC) transports heat over large distances and is therefore an essential component of the Earth's climate system, both today and in the past (Ganopolski and Rahmstorf, 2001; Gordon, 1986; Rahmstorf, 1996; Stute et al., 2001). It is a major focus of paleoceanography to understand the contribution of the AMOC to glacial-interglacial climate change (Boyle and Keigwin, 1987; Broecker and Hemming, 2001; Clark et al., 2002; Knorr and 25  Lohmann, 2003; Knorr et al., 2021).

The state of deep-water formation during the Last Glacial Maximum (LGM) has been discussed in the paleoclimate literature. Based on water mass properties, North Atlantic Deep Water (NADW) formation was shallower (Butzin et al., 2005; Curry and Oppo, 2005; Duplessy et al., 1988; Ferrari et al., 2014; Hesse et al., 2011; Lippold et al., 2012; Lund et al., 2011; Lynch-Stieglitz et al., 2007; Muglia et al., 2018; Skinner et al., 2017), but not much weaker (McManus et al., 2004; Sarnthein 30  et al., 1994). At the same time, Antarctic Bottom Water (AABW) export from the Southern Ocean increased (Ledbetter and

Johnson, 1976; Negre et al., 2010; Robinson et al., 2005). The salinity of glacial AABW could have been much greater than today, leading to an enhanced stratification of the glacial ocean between the upper and lower cells (Adkins et al., 2002; Bouttes et al., 2009; Francois et al., 1997; Jansen, 2017; Klockmann et al., 2016; Knorr et al., 2021; Lund et al., 2011; Stein et al., 2020; Watson and Garabato, 2006). Several modeling studies provide a physical basis for the shoaled glacial AMOC, likely

to be caused by changes in Southern Ocean sea ice (Baker et al., 2020; Butzin et al., 2005; Ferrari et al., 2014; Jansen and Nadeau, 2016; Marzocchi and Jansen, 2017; Nadeau et al., 2019; Sun et al., 2018; Sun et al., 2020; Watson et al., 2015) or terrestrial ice input (Miller et al., 2012).

Coupled ocean-atmosphere model simulations of the LGM climate reveal a broad spectrum of results, showing considerable disagreement regarding whether the AMOC was weaker or stronger compared to present-day (PD) conditions

(Kageyama et al., 2021; Knorr et al., 2021; Otto-Bliesner et al., 2007; Weber et al., 2007; Zhang et al., 2013). However, a critical factor often overlooked in these analyses is the significantly enhanced tidal dissipation during the LGM (Arbic et al., 2004b; Egbert et al., 2004; Green, 2010; Griffiths and Peltier, 2008; Griffiths and Peltier, 2009; Wilmes and Green, 2014). Incorporating this element into the models, as demonstrated in the research by Schmittner et al. (2015) and Wilmes et al. (2019), leads to a notable finding: both the depth and the strength of the AMOC during the LGM are substantially increased

when the changes in tidal dissipation are taken into account. This suggests a pivotal role of tidal mixing in shaping the LGM's ocean circulation dynamics.

Currently, tides provide about half, or 1 TW (1 TW = $10^{12}$ W), of the energy to maintain the global meridional overturning circulation (MOC) (Ferrari and Wunsch, 2009; Wunsch and Ferrari, 2004). Numerous studies have suggested a significant intensification of tides due to the 120-130 m drop in global mean sea level and exposure of continental shelves during the

LGM (Arbic et al., 2004b; Egbert et al., 2004; Green, 2010; Griffiths and Peltier, 2008; Griffiths and Peltier, 2009; Wilmes and Green, 2014). This exposure reduces the effective damping, leading to an increase in tides. Additionally, there is greater tidal dissipation in the deep ocean interior rather than on the continental shelves during the LGM. This amplified tidal dissipation may have been a critical factor in driving a more vigorous glacial AMOC compared to current levels, as postulated by Green et al. (2009), Schmittner et al. (2015), and Wilmes et al. (2019). Therefore, changes in tidal dissipation do play an

important role and should not be neglected in paleoclimate simulations (Schmittner et al., 2015). It is noteworthy that Wilmes et al. (2021) achieves a relatively shoaled AMOC through the artificial reduction of meridional moisture flux and precipitation at high latitudes. However, to date, no research has directly demonstrated a shoaled AMOC under a realistic LGM forcing conditions, despite the presence of enhanced glacial tidal dissipation.

The primary objectives of this study are threefold: (1) to reproduce a stratified glacial ocean and a shoaled AMOC under

actual LGM forcing conditions, considering both increased glacial local and far-field tidal dissipation, (2) to analyze the reasons for a shoaled AMOC despite the presence of enhanced tidal dissipation during the LGM, and (3) to confront modeled ocean circulation with paleoclimate reconstructions. To achieve these goals, we employ a global ocean general circulation model to generate a series of ocean circulation scenarios. These scenarios are driven by both LGM and PD surface forcing, as well as varying degrees of tidal mixing. Previous studies have already explored the role of increased glacial ocean stratification

in causing a shallower AMOC during the LGM (Jansen and Nadeau, 2016; Jansen, 2017). In this context, our analysis underscores that despite the nearly threefold intensification of tidal dissipation during the LGM, enhanced stratification still plays a dominant role in maintaining a shoaled glacial AMOC.

## 2 Materials and Methods

### 2.1 Tidal model

The global tidal model is based on the Finite Volume Community Ocean Model (FVCOM), which is based on an unstructured finite-volume model with triangular meshes (Chen et al., 2003). The tidal model solves the equations

$$\frac{\partial \mathbf{u}}{\partial t} + f \times \mathbf{U} + \mathbf{U} \cdot \nabla \mathbf{u} = -gH\nabla\left(\zeta - \alpha\zeta_{EQ} - \zeta_{SAL}\right) - a_H\nabla^2\mathbf{U} + \mathbf{D}_{BL} + \mathbf{D}_{IT}, \tag{1}$$

where u is the horizontal velocity, $\mathbf{U} = \mathbf{u}H$ is the horizontal transport speed, $f$ is the Coriolis parameter, $g$ is the gravitational acceleration, $\zeta$ is the instantaneous tide level; $\zeta_{EQ}$ is the equilibrium tide level (Hendershott, 1972); α is the body tide Love

number; $\zeta_{SAL}$ is the gravitational self-attraction and loading tide term, which has been implemented using an iterative method (Arbic et al., 2004a; Egbert et al., 2004). $A_H$ is the horizontal turbulent eddy viscosity coefficient. Momentum is dissipated through two processes: first, a quadratic (in velocity) bottom friction term

$$\mathbf{D}_{BL} = -C_d\mathbf{u}|\mathbf{u}|, \tag{2}$$

in which the bottom friction coefficient $C_d$ is taken as 0.0025. Second, $\mathbf{D}_{IT}$ is the internal wave drag, the linear transfer of

energy to internal waves, based on Zaron and Egbert (2006):

$$\mathbf{D}_{IT} = \Gamma H(\nabla H)^2 \frac{N_b\bar{N}}{8\pi^2\omega}\mathbf{u}. \tag{3}$$

Here, $\Gamma = 50$ is a scaling factor. $N_b$ and $\bar{N}$ are buoyancy frequency at the seafloor and depth-averaged vertical value, respectively, both derived from our Finite-volumE Sea ice-Ocean Model 2.0 (FESOM 2.0, see description in section 2.2) simulations. It is noteworthy that the tidal dissipation obtained from the tidal model may further influence the $N^2$ value obtained

in FESOM 2.0, leading to certain sensitivities. We have employed an iterative process to eliminate these sensitivities, with the detailed iterative process provided in Appendix A1. $\omega$ is the tidal frequency of the M2 tide. The tidal model utilizes four major tidal constituents forcing (M2, S2, K1, and O1), accounting for more than 94 % of today's dissipation (Egbert and Ray, 2003). The experiments are executed for a total of 30 days, with the final 20 days used for harmonic analysis.

The resolution of the model ranges from 10 km to 40 km, with higher resolution in the shallow waters and areas with

significant water depth changes. For the PD triangular mesh, the node number is 422,932, and the cell number is 817,641. For the LGM, the numbers are 323,101 and 624,641, respectively. The term "node" refers to the vertices of the unstructured triangular mesh, whereas "cell" denotes the triangles formed by connecting these nodes.

Here, we calculate the bottom friction dissipation $D_{BL}$ and the internal-tide dissipation $D_{IT}$ due to linear transfer of energy to internal waves:

$D_{IT} = < \rho_0\mathbf{u} \cdot \mathbf{D}_{IT} >, D_{BL} = < \rho_0\mathbf{u} \cdot \mathbf{D}_{BL} >, \tag{4}$

where we set the reference density to 1,035 kg/m³, and brackets <> denote the tide period of the respective tidal constituent.

Figure 1 presents the bottom friction dissipation ($D_{BL}$) and internal-tide dissipation ($D_{IT}$) during the PD and the LGM. A significant difference in their distributions is observed: bottom friction dissipation ($D_{BL}$) is mainly concentrated in shallow sea areas, while internal-tide dissipation ($D_{IT}$) is primarily found in deep-sea regions with notable topographic variations, such as mid-ocean ridges. Moreover, compared to the PD, the primary regions of energy dissipation during the LGM shifted from shallow to deep seas. During the PD, dissipation below 500 meters depth accounted for 1.86 TW, which is 62% of the total dissipation. In contrast, during the LGM, dissipation below 500 meters is 1.35 TW, accounting for only 29% of the total dissipation (Table 1). These data are consistent with previous research findings (Arbic et al., 2004a; Egbert et al., 2004; Green, 2010; Griffiths and Peltier, 2008; Griffiths and Peltier, 2009; Wilmes and Green, 2014).

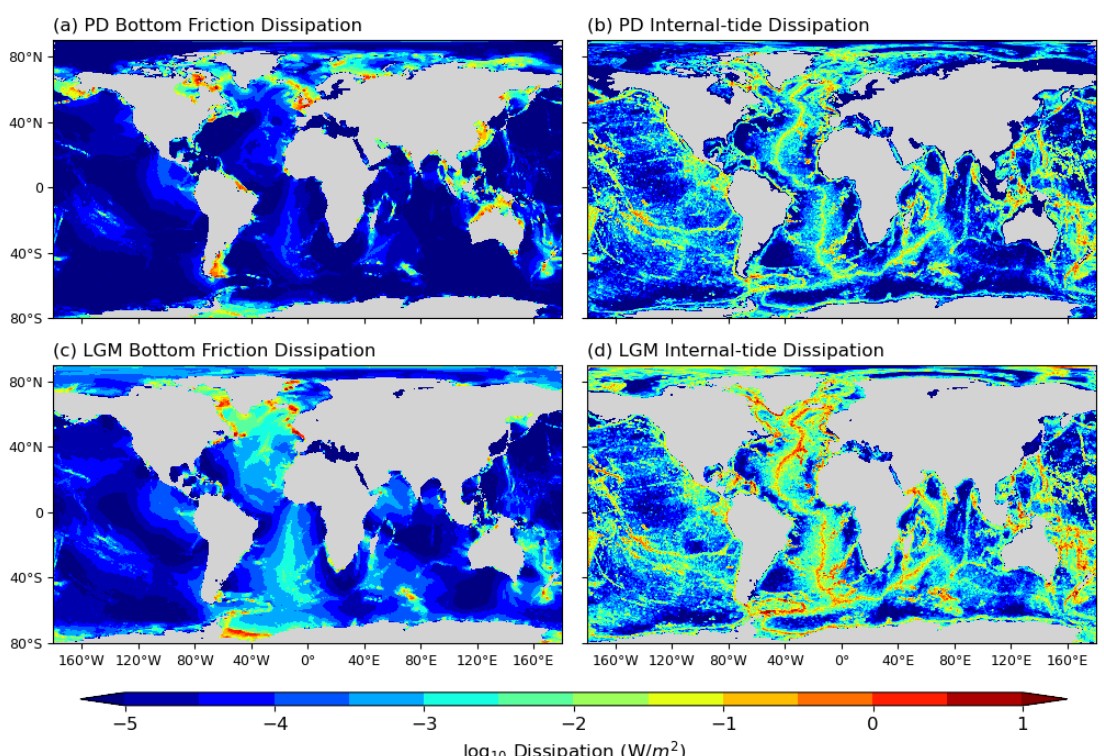

**Figure 1.** Global distributions of bottom friction dissipation ($D_{BL}$) and internal-tide dissipation ($D_{IT}$) during PD and LGM.

**Table 1.** Global and Sub-500m Distribution of $D_{IT}$ and $D_{BL}$ during PD and LGM (TW).

| Time | $D_{IT}$ | $D_{IT}$ (<500 m) | $D_{BL}$ | $D_{BL}$ (<500 m) | $D_{Total}$ | $D_{Total}$ (<500 m) |
|------|----------|-------------------|----------|-------------------|-------------|----------------------|
| PD | 1.31 | 0.21 | 1.69 | 1.65 | 3.00 | 1.86 |
| LGM | 3.41 | 0.36 | 1.17 | 0.99 | 4.58 | 1.35 |

### 2.1.1 Bathymetry

The PD bathymetry comes from the 1-min RTOPO2 database (Schaffer et al., 2016). For the LGM bathymetry, we use sea level from the ICE-5G (VM2 L90) version 1.2 (Peltier, 2004). Notably, the tidal dissipation derived using ICE-6G is actually weaker than that obtained using ICE-5G (Wilmes et al., 2019; Wilmes et al., 2021). Here, we have chosen to use ICE-5G to investigate whether the AMOC during the LGM would be affected under these stronger tidal conditions. The sea-level difference between PD and LGM is calculated by subtracting the PD from the LGM sea levels in the respective ICE-5G dataset. The low-resolution paleo sea level changes (1° horizontal resolution) are then interpolated to the grid of RTOPO2 and added to PD RTOPO2 bathymetry in order to retain the high-resolution topographic features. Finally, we interpolate the high-resolution bathymetry to the unstructured triangular mesh of the tidal model.

### 2.1.2 Tide model validation

The harmonically analyzed amplitudes (complex notation sinusoids) are used to evaluate the elevations. Simulated sinusoids $\hat{\zeta}$ were interpolated to the 1/6 ° grid of TPXO9.v1 and compared to the reference sinusoids $\widehat{\zeta_R}$ by evaluating the spatially averaged root mean square (RMS) error $\Delta\zeta$

$$\Delta\zeta = \sqrt{\frac{\iint |\hat{\zeta} - \widehat{\zeta_R}|^2 \, dA}{2 \iint dA}}. \tag{5}$$

The RMS errors were calculated for four tidal constituents and separately for deep water regions (depths > 500 m) and shallow shelf seas. The results are presented in Table 2. Meanwhile, Table 3 provides a comparison of the M2 tide RMS with other forward tidal models.

**Table 2.** Tide model RMS Error for four tidal constituents (Units: cm)

|  | Deep water regions | Shallow shelf seas |
| --- | --- | --- |
| M2 | 4.87 | 15.14 |
| S2 | 1.85 | 6.34 |
| K1 | 1.67 | 5.23 |
| O1 | 1.31 | 3.71 |

**Table 3.** Comparison of M2 RMS error in Forward Tide Models

| Model | Deep Water (cm) | Shallow Water (cm) | Global (cm) |
|---|---|---|---|
| Our Tidal Model | 4.87 | 15.14 | 6.54 |
| Egbert et al. (2004) | < 5 | NA | NA |
| Arbic et al. (2004a) | 7.26 | NA | NA |
| Griffiths and Peltier (2009) | NA | NA | 13.6 |
| Wilmes and Green (2014) | 3.86 | NA | 6.67 |
| Schindelegger et al. (2018) | 4.4 | 14.6 | NA |

*Note: Differences in the observed models selected across different studies may influence the results.

## 2.2 Ocean model

The FESOM 2.0 (Danilov et al., 2017), which is the ocean component of the AWI Earth System Model (Sidorenko et al., 2019), is employed in our experiments. FESOM 2.0 solves the primitive equations in the Boussinesq and hydrostatic approximations. It adopts an unstructured triangular mesh framework, with scalar degrees of freedom located at vertices and horizontal velocities at triangle centers. Additionally, the Finite Element Sea Ice Model (Danilov et al., 2015) is incorporated into FESOM 2.0 as a set of subroutines. FESIM solves the modified elastic-viscous-plastic (mEVP) dynamical equations, enabling a reduction in subcycling steps while maintaining numerical stability (Kimmritz et al., 2017; Koldunov et al., 2019). Figure 2 presents the horizontal resolution of the PD and LGM mesh configurations used in this study. The PD mesh consists of 126,858 nodes and 244,659 cells, while the LGM mesh comprises 104,425 nodes and 203,142 cells. Both meshes have the same nominal resolution of 1° in most parts of the global ocean, approximately 25 km north of 50° N, about 1/3° at the Equator, and 10 km for the Arctic Ocean and Bering Sea.

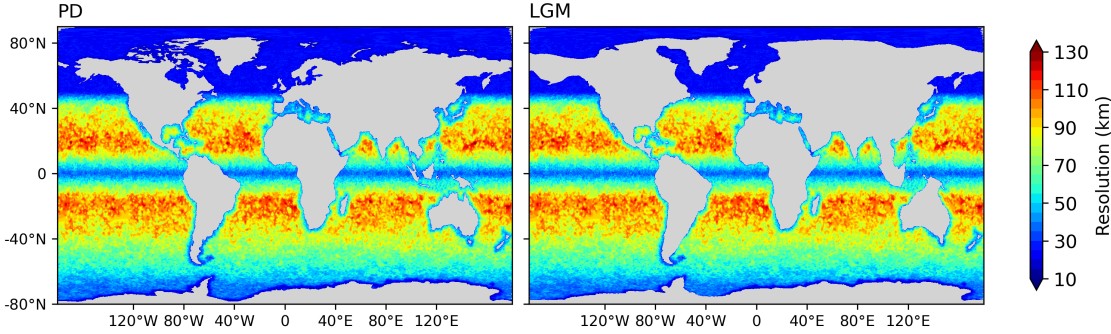

**Figure 2.** Horizontal resolution of PD (left) and LGM (right) mesh configurations used in this study.

The K-profile parameterization (Large et al., 1994) is utilized universally, targeting surface ocean mixing, whereas a constant vertical background diffusivity $k_{bg}$ is employed to manage the effects of various background mixing mechanisms.

In FESOM 2.0, the default value for $k_{\mathrm{bg}}$ is set to $0.1 \times 10^{-4}$ m$^2$/s. Furthermore, the tidal mixing parameterization by Schmittner and Egbert (2014), drawing on the foundational work of Jayne and St. Laurent (2001) and Simmons et al. (2004), is incorporated. This parameterization uniquely accounts for the influence of subgrid-scale bathymetry on the penetration depth of energy inputs and differentiates between diurnal and semidiurnal tidal effects. The tidal diapycnal diffusivity, $k_{v\_tidal}$, is

given by

$$k_{v\_tidal} = \frac{\Gamma \epsilon}{N^2}. \tag{6}$$

$\Gamma$ is the mixing efficiency which is set to 0.2 and $N^2$ is the buoyancy frequency. The rate of tidal energy dissipation, $\epsilon$, is

$$\epsilon = \frac{1}{\rho} \sum_{z'>z}^{H} \sum^{\mathrm{TC}} q_{\mathrm{TC}} D_{\mathrm{IT,TC}}(x,y) F(z,z'), \tag{7}$$

where $D_{\mathrm{IT,TC}}(x,y)$ is the internal-tide energy flux from barotropic tides to the internal tides from the tidal model, and $F$ is the

vertical decay function using an e-folding depth of 500 m above the seafloor $H$. The local dissipation efficiency $q_{\mathrm{TC}}$, accounts for the critical latitude $y_c$ of diurnal and semidiurnal tidal constituents (TC)

$$q_{\mathrm{TC}} = \begin{cases} 1, \text{for } |y| > y_{c,\mathrm{TC}} \\ 0.33, \ \text{otherwise} \end{cases}, \tag{8}$$

$y_c$ is 30° for the diurnal constituents (K1 and O1) and 72° for the semidiurnal constituents (M2 and S2).

### 3 Model and Experiments

In the Methods section, we extract the internal-tide dissipation, denoted as $D_{IT}$, from the global tide model. In comparison to the PD values, the $D_{IT}$ of the four principal tidal constituents (M2, S2, K1, and O1) during the LGM shows a nearly threefold increase, escalating from 1.31 TW to 3.41 TW (Table 1). The predominant contributor to this shift is the M2 tide, with a period of 12.42 hours, which closely matches the North Atlantic basin's period of 12.66 hours (Muller, 2008), creating resonance. During the LGM, the removal of continental shelves decreased damping, causing a significant increase in

the M2 tide. Its value surged from a PD level of 0.89 TW to reach 2.94 TW during the LGM. These values are in close agreement with previous research findings (Arbic et al., 2004a; Egbert et al., 2004; Green, 2010; Griffiths and Peltier, 2008; Griffiths and Peltier, 2009; Wilmes and Green, 2014). The horizontal distributions of $D_{IT}$ (Figure 1) are used as input to a tidal mixing parameterization in FESOM 2.0.

The experiments are designed to explore how tidal mixing impacts the glacial AMOC, with specifics outlined in Table 4.

PD simulations are forced by the 1958-2020 period in the Reanalysis dataset (JRA55-do 1.4.0) by Kobayashi et al. (2015): it represents the second global atmospheric reanalysis conducted by the Japan Meteorological Agency (JMA). Spanning from 1958 onwards, it employs the TL319 version of JMA's operational data assimilation system as of December 2009. JRA-55 addresses several issues identified in previous reanalyses and enhances the temporal consistency of temperature analysis, thereby making it suitable for examining multidecadal variability and conducting climate simulations. We repeatedly

conducted each PD case simulation using the 1958-2020 JRA-55 data five times to achieve simulation stability. No significant

trend was detected, and we utilized the average results from the final cycle for our analysis. The simulations conducted for the PD scenario using FESOM 2.0 have been thoroughly validated. Detailed assessments and descriptions of the PD case's configuration can be found in Scholz et al. (2019) and Scholz et al. (2022). Figure 3 depicts a comparison of the temperature and salinity in the Atlantic Ocean from our PD case with the WOA (World Ocean Atlas) 2018 data. The results indicate that

our model accurately reproduces the temperature and salinity structures of the modern ocean. This lays a solid foundation for further simulations of the LGM ocean and the study of the role of tides.

Regarding the LGM simulations, the differences in model configuration are attributed to surface forcing and initial conditions. Both of these two are taken from the LGMW case in Zhang et al. (2013). It is worth noting that selecting an appropriate forcing for the LGM simulations is crucial. In Knorr et al. (2021), a comparison of the LGM simulations of Atlantic

Ocean temperature and salinity structure was conducted among different models from the Paleoclimate Model Intercomparison Project (PMIP, Braconnot et al., 2007; Weber et al., 2007) and Coupled Model Intercomparison Project (CMIP, Braconnot et al., 2012; Kageyama et al., 2017). From these, we selected the simulation results of the well-performing climate model COSMOS (Zhang et al., 2013) as the surface forcing for the LGM simulations in this study. This provides a solid foundation for accurately simulating the glacial ocean. The LGM simulations are executed over a duration of 600 years to achieve a quasi-

equilibrium state. A time series depicting the strength of AMOC for the LGM cases is presented in Figure S1. The concluding 62 years of this period were selected. The simulations are summarized in Table 4.

**Table 4.** Experimental design of the simulations in this study.

| Simulation | Surface forcing | Tidal mixing | Initial conditions |
|---|---|---|---|
| PD | PD | No | PD |
| PD_tidal | PD | PD | PD |
| PD_glacial_tidal | PD | LGM | PD |
| LGM | LGM | No | LGM |
| LGM_tidal | LGM | LGM | LGM |

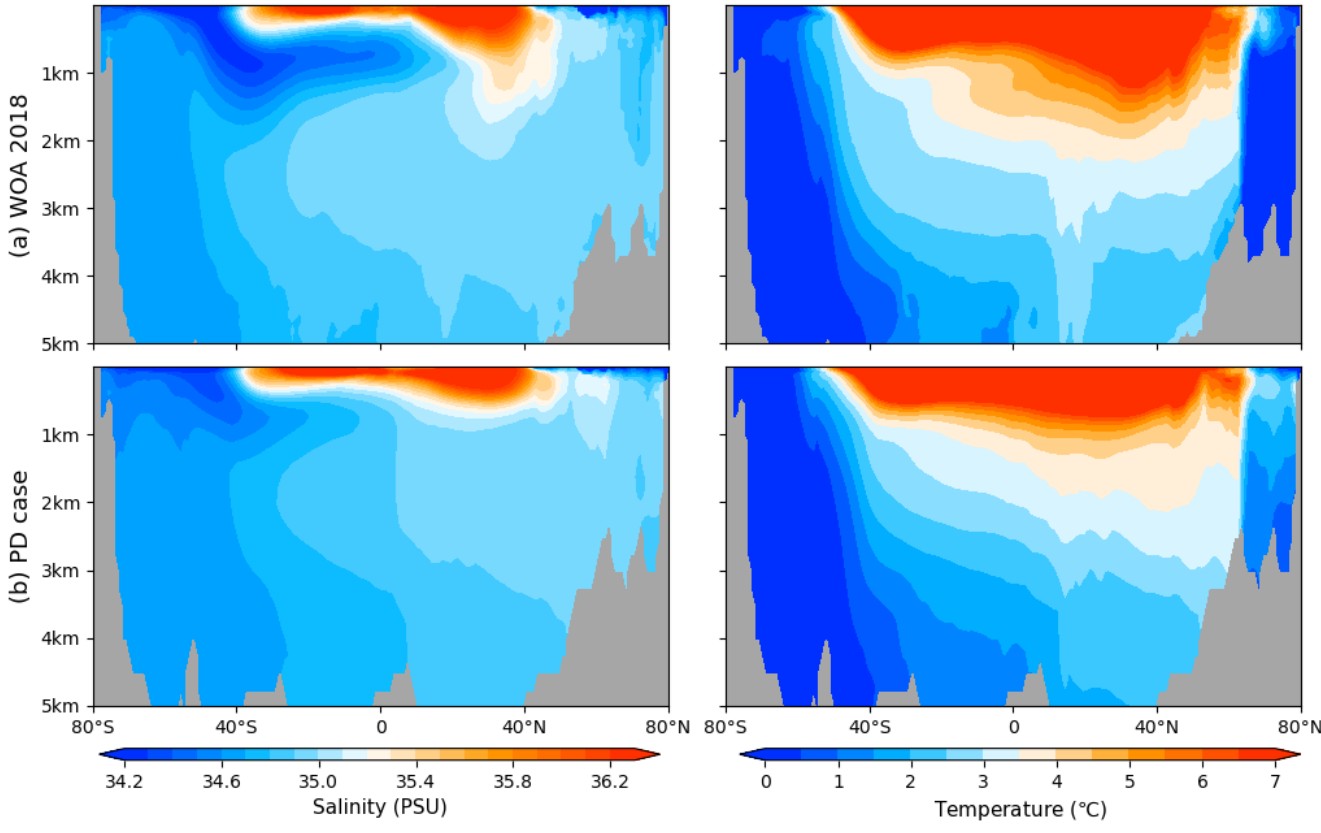


**Figure 3.** Comparison of salinity and temperature between WOA 18 data and PD simulation in the Atlantic Ocean.

## 4 Results

The AMOC strength varies within a range of 11.6 to 15.5 Sv for the PD and 13.3 to 13.9 Sv for the LGM, respectively
(Figure 4, Table S1). It is noteworthy that both LGM simulations (LGM and LGM_tidal) exhibit a shoaled AMOC, identified at approximately 1700 meters depth. This suggests that the inclusion of tidal mixing does not affect the glacial AMOC's configuration. Instead, stratification is identified as a crucial factor contributing to the shoaled AMOC, which is more pronounced in the upper and middle glacial ocean. Stratification, on one hand, signifies the buoyancy forces encountered by the water masses during their descent. On the other hand, it exerts a notable influence on the model tidal diffusivities, as
outlined in equation (6). Our tidal model results indicate that internal-tide dissipation $D_{IT}$ in the North Atlantic during the LGM reached 0.81 TW, a significant increase compared to the 0.13 TW observed in the PD, representing a sixfold increase. Additionally, the average squared buoyancy frequency $N^2$ at depths of 1000-3000 meters in the Atlantic during the LGM is $1.89 \times 10^{-6}$ s$^{-2}$, compared to $1.02 \times 10^{-6}$ s$^{-2}$ in the PD. These changes do not suggest tidal dominance but rather imply stronger

vertical tidal diffusivity (shown in Figure 4b, e). This enhanced vertical mixing aligns with the observed reductions in the
vertical gradients of radiocarbon and $\delta^{13}C$ in the deep Atlantic during the LGM (Skinner et al., 2017; Muglia et al., 2018; Peterson et al., 2014; Molina-Kescher et al., 2016; Sikes et al., 2016).

However, the incorporation of tidal mixing processes results in a substantial increase in the generation of AABW, with a magnitude of -7.9 Sv, significantly exceeding the PD estimate of -4.2 Sv. This enhancement is in agreement with the results derived from paleo-proxy data, which indicate an intensified glacial Atlantic AABW (Curry and Oppo, 2005; Zhang et al.,
2017). Consequently, while LGM tides may not modify the AMOC, they exert a significant influence on AABW formation. Thus, accounting for tidal effects remains essential in climate modeling for the LGM period.

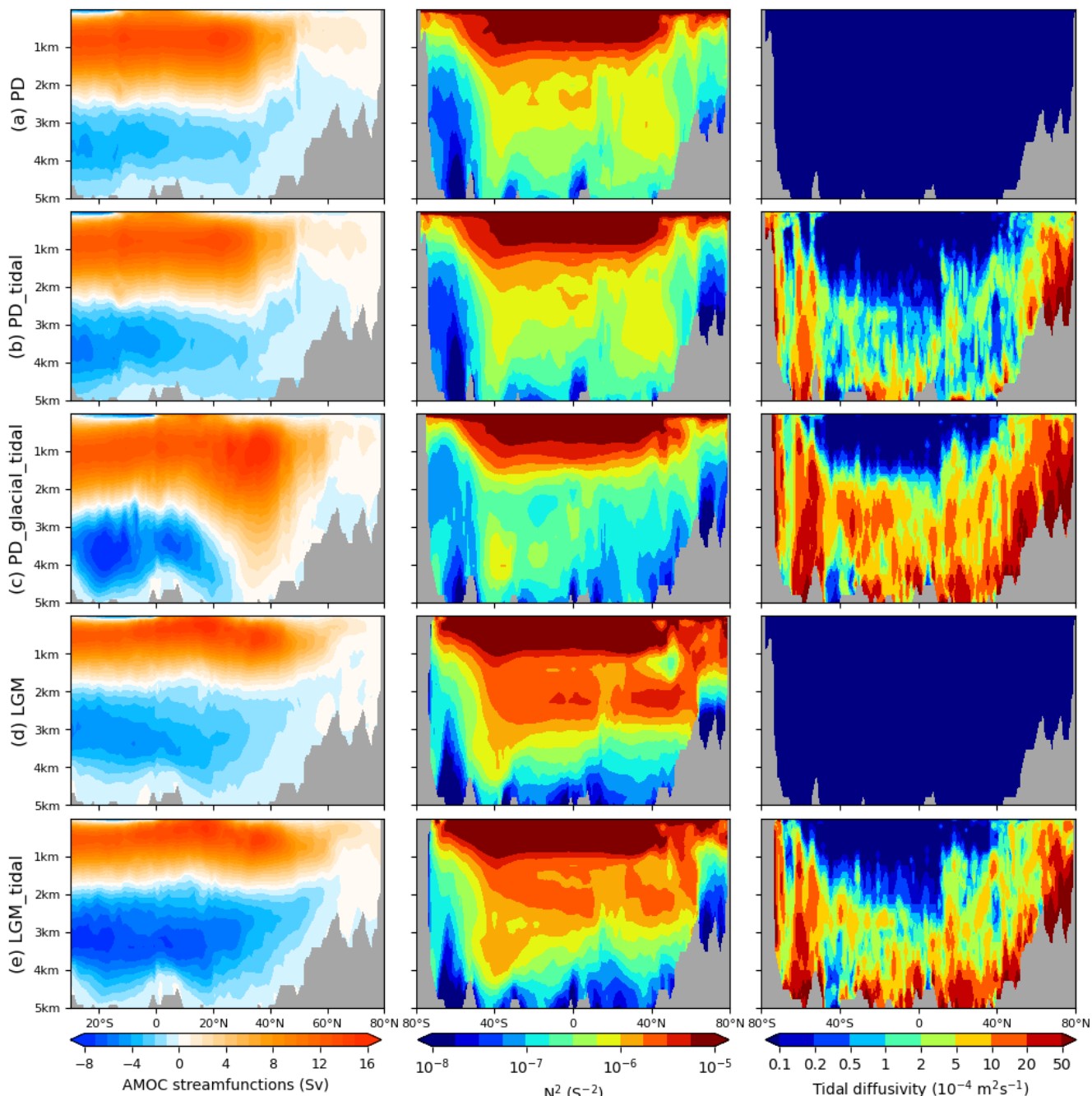

**Figure 4.** AMOC (left) and zonally averaged distributions for the squared buoyancy frequency (middle), and tidal diffusivity (right) in the Atlantic Ocean. Simulations as listed in Table 4.


In the PD_tidal experiment, incorporating the tidal mixing parameterization does not alter the geometry of the AMOC. However, the PD_glacial_tidal simulation, which includes LGM tidal dissipation, reveals distinct dynamics. This simulation demonstrates a significant increase in both the strength and depth of the AMOC, extending to near-benthic layers at approximately 35°N (Figure 4c and Table S1), and is accompanied by notably reduced stratification. These observations suggest that the effects and dynamics of enhanced tidal dissipation differ substantially under the varying ocean stratification intensities during the LGM and PD periods.

To investigate the origins of various stratifications from the perspectives of temperature and salinity, we conduct a further analysis of the temperature and salinity distributions in the Atlantic Ocean across different cases (Figure 5). Under PD forcing, the surface salinity below 40 degrees latitude is significantly higher than in the mid and lower layers, which weakens ocean stratification. This is in stark contrast with the abyssal high salinity observed during the LGM, a defining feature of the glacial ocean (Adkins et al., 2002; Knorr et al., 2021). Regarding temperature, a decrease from the surface to the seabed is observed in both the PD and the LGM scenarios. Notably, during the LGM, the simulated temperature exhibits a pronounced decrease above a depth of 2 km, descending to 0°C at this level. Beneath 2 km, the ocean is relatively homogeneous and close to the freezing point, indicating a cold and well-mixed deep ocean, consistent with paleo-proxy data (Adkins et al., 2002). This steeper temperature gradient enhances the stratification in the upper layers of the glacial ocean. Collectively, the abyssal high salinity alongside the swift vertical temperature decline significantly contributes to a more pronounced stratification within the glacial ocean. Accurately replicating these temperature and salinity features is crucial in the climate modeling of the LGM.

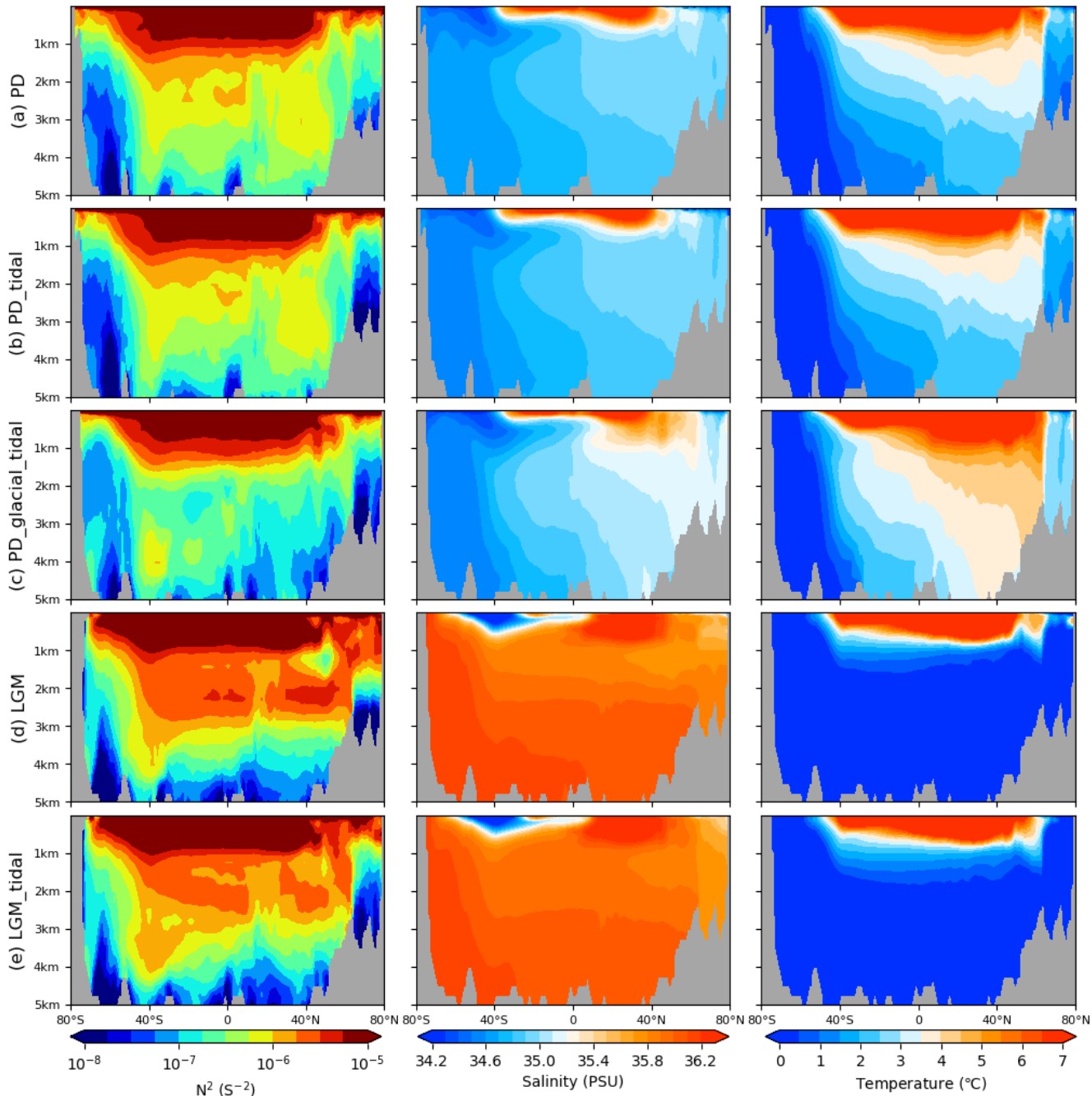

**Figure 5.** Zonally averaged distributions in the Atlantic Ocean for the simulations listed in Table 4: Squared buoyancy frequency (left), salinity (middle), and potential temperature (right).

Regarding our similar consideration of enhanced tidal mixing during the LGM, the reason our results differ from those of previous studies (Schmittner et al., 2015; Wilmes et al., 2019) is believed to be because Schmittner et al. (2015) and Wilmes

et al. (2019) do not reproduce the high abyssal salinity and enhanced stratification in the LGM Atlantic, which have an opposite effect on the AMOC compared to the stronger tidal mixing.

## 4 Discussion

Tides play a pivotal role in climate dynamics, such as facilitating the release of iceberg armadas during Heinrich events (Arbic et al., 2004b), and serving as the primary driving force behind both vertical and horizontal ice sheet movements at their marine peripheries (Padman et al., 2018). Focusing on the LGM period reveals that glacial tidal dissipation was approximately three times greater than present levels. This increase, combined with the closure of the Bering Strait (Hu et al., 2010), led to reduced freshwater transport to the Atlantic, ostensibly resulting in a strengthened glacial AMOC. However, paleoclimatic proxy data indicate a significant shoaling of the AMOC during the LGM, with an estimated reduction in depth of about 1,000 m compared to contemporary conditions (Burke et al., 2015; Lund et al., 2011). The primary aim of this study is to identify the reasons for the shoaled glacial AMOC, given the complex interplay of these factors.

Our results indicate that the integration of the additional tidal mixing parameterization does not significantly influence the AMOC in either the PD or LGM. In the PD scenario, the relatively weak tides can be adequately accounted for by background diffusivity $k_{bg}$, thereby negating the necessity for an additional tidal parameterization. Furthermore, during the LGM, tides are unlikely to play a substantial role in influencing the glacial AMOC due to pronounced ocean stratification. This stratification hampers the mixing of water masses, on one hand, and, on the other, it leads to a decrease in the effectiveness of tidal mixing. This is because the buoyancy frequency, which appears in the denominator in the parameterization of tidal mixing (as detailed in equation (6) in the Method section), suggests that stronger stratification significantly reduces the impact of tidal dissipation. However, in the abyssal ocean with relatively weak stratification, the pronounced tidal dissipation during the LGM notably enhances the formation of AABW.

The tidal mixing parameterization considers only locally dissipated energy, which accounts only for one-third of the total energy (Jayne, 2009; Schmittner and Egbert, 2014). The remaining two-thirds of the energy is dissipated in the far-field, where the background diffusivity $k_{bg}$ is employed to represent this dissipation. Consequently, we calculated the far-field dissipation due to $k_{bg}$ and the local tidal dissipation due to $k_{v\_tidal}$ for each simulation using the Osborn (1980) formula:

$$P = \int \rho \epsilon \ dV = \frac{1}{\Gamma} \int \rho k \, N^2 \, dV.$$

The results are presented in Table 5. Additionally, we conducted another experiment, LGM_tidal_3, in which $k_{bg}$ increased from $1*10^{-5}$ m²s⁻¹ to $3*10^{-5}$ m²s⁻¹ for comparison. The results indicate that the LGM_tidal experiment underestimates the total energy contribution from tides by only 2.05 TW, whereas LGM_tidal_3 reaches 3.92 TW.

Figure 6 presents the AMOC geometry for LGM_tidal and LGM_tidal_3. The geometry of the AMOC in LGM_tidal_3 remains relatively shallow without significant changes, which further supports our study's conclusions. The only notable change is in the AMOC strength, which increased from 13.3 Sv to 15.4 Sv. This underscores the necessity of employing the tidal mixing parameterization and also the importance of appropriately adjusting the background diffusivity ($k_{bg}$). Additionally,

this result is the first to demonstrate that the glacial shallower AMOC geometry during the LGM remains unchanged, even when considering far-field tidal dissipation.

**Table 5.** Summary of energy consumption due to diapycnal mixing

| Simulation | $k_{bg}$ ($1*10^{-5}$ m$^2$s$^{-1}$) | Far-field contribution (TW) | Local contribution (TW) | Total (TW) |
|---|---|---|---|---|
| PD | 1 | 0.79 | 0 | 0.79 |
| PD_tidal | 1 | 0.78 | 0.38 | 1.16 |
| LGM | 1 | 1.05 | 0 | 1.05 |
| LGM_tidal | 1 | 1.02 | 1.03 | 2.05 |
| LGM_tidal_3 | 3 | 2.86 | 1.06 | 3.92 |

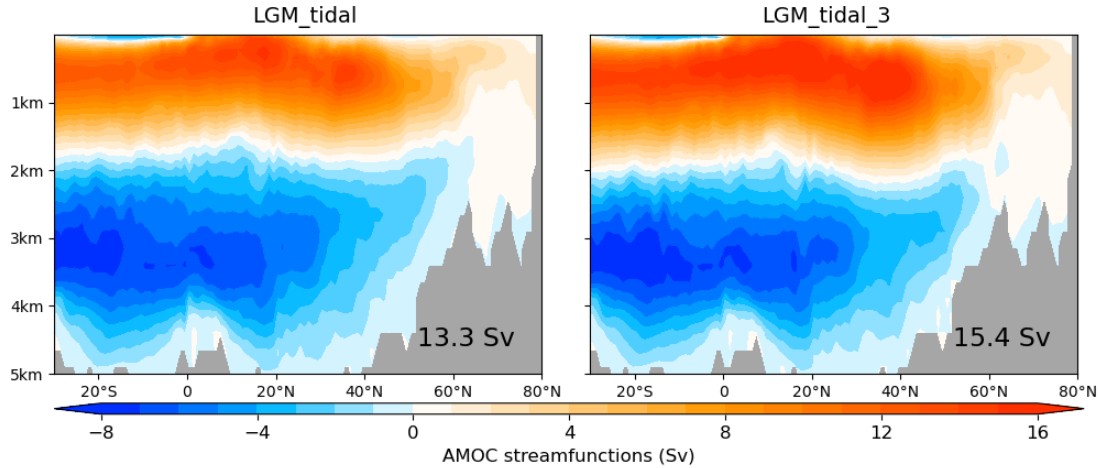


**Figure 6.** AMOC streamfunctions (Sv) for LGM_tidal and LGM_tidal_3.

The application of enhanced LGM tidal dissipation to PD conditions (in the case of PD_glacial_tidal), where ocean stratification is significantly weaker than LGM, yields entirely different results. In this scenario, the amplified tidal dissipation induces a deeper and more potent AMOC. We propose a potential positive feedback mechanism accounting for the increased
AMOC during termination. A reduced ocean stratification during the initial phase of termination enhances the effectiveness of tidal mixing, a process analogous to the one discussed above. This increased tidal mixing will affect the ocean more efficiently, further weakening ocean stratification, which in turn increases tidal mixing. This initiates a positive feedback loop that culminates in reduced stratification and a more vigorous and deeper AMOC.

**5 Conclusions**

The concept of enhanced glacial ocean stratification, potentially resulting from cooling and salinification of glacial AABW, which could lead to a shoaled AMOC during the LGM has previously been discussed (Jansen and Nadeau, 2016;

Jansen, 2017; Klockmann et al., 2016). However, until now, no research has directly demonstrated a shoaled AMOC under real LGM forcing conditions, including the impact of increased glacial tidal dissipation. Montenegro et al. (2007) proposes that LGM tides have a minimal impact on the AMOC, attributing this to a potential underestimation of tidal dissipation during the LGM. In contrast, Schmittner et al. (2015) and Wilmes et al. (2019) suggests a significant enhancement and deepening of the North Atlantic overturning cell under vigorous glacial tidal dissipation. It is noteworthy that Wilmes et al. (2021) obtained a relatively shoaled LGM AMOC through the artificial reduction of meridional moisture flux and precipitation at high latitudes.

Our study is the first to directly demonstrate a stratified ocean and shoaled AMOC under a real LGM forcing conditions without any artificial modifications, despite the context of increased glacial tidal dissipation. We suggest that accurately simulating the high salinity of the deep sea and the rapid temperature changes in the ocean's upper layers is crucial for correctly reproducing a glacial stratified ocean. In such an environment, the significant tidal dissipation during the LGM was insufficient to counter the increased ocean stratification, leading to a shoaled AMOC. Furthermore, we emphasize that this notable glacial tidal dissipation plays a critical role in strengthening the AABW during the LGM.

Our results highlight the dominance of background conditions and mixing on ocean circulation dynamics, with possible complex feedbacks in the Earth system (Lohmann et al., 2020). Here, we use an ocean-only model, which means the LGM atmospheric forcing is kept constant. Consequently, this approach has a limitation: it does not account for interactions between the ocean or sea ice and the atmosphere. As a logical next step, we incorporate tidal energy dissipation into fully coupled Earth system models (Zhang et al., 2014; Liu et al., 2009) to elaborate AMOC dynamics during deglaciation.

**Appendix A: Detailed Iterative Process for Eliminating $N^2$ Sensitivities between the Tidal Model and FESOM 2.0**

To address the $N^2$ sensitivities and interactions between FESOM2.0 and the tidal model, we employed an iterative process for PD_tidal and LGM_tidal simulations. Here, we provide a detailed description of this process, using the LGM_tidal simulation as an example.

Iterative Process Steps:

1. Initial Input: We begin by obtaining the $N^2$ from the LGM simulation which is without tidal mixing, and this initial $N^2$ is then used as input for the tidal model.

2. First Iteration (LGM_tidal1): The tidal model, using the initial $N^2$, calculates the tidal dissipation, which is then fed back into the FESOM2.0, producing the first experimental result, termed LGM_tidal1.

3. Second Iteration (LGM_tidal2): The $N^2$ from the LGM_tidal1 simulation is input into the tidal model again to generate a new tidal dissipation, which is incorporated into the FESOM2.0, and the model is run to obtain LGM_tidal2.

4. Final Output (LGM_tidal2): The LGM_tidal2 simulation represents the second iteration and is the LGM_tidal experiment presented in our manuscript.

340   The Figure A1 illustrates the changes in depth-averaged vertical $N^2$ during these iterations for both the PD and LGM scenarios, showing that for the LGM simulations, the primary change from the initial LGM to LGM_tidal1 involves a decrease in $N^2$ in the Arctic, and from LGM_tidal1 to LGM_tidal2, there is minimal change observed, whereas for the PD simulations, there were no significant changes in $N^2$ throughout the iterations, thus effectively minimizing the mutual influences between $N^2$ in the tidal model and the OGCM after the iterative process. This iterative approach ensures the

345   stability and accuracy of our model results, reducing the sensitivity of $N^2$ to tidal dissipation feedbacks.

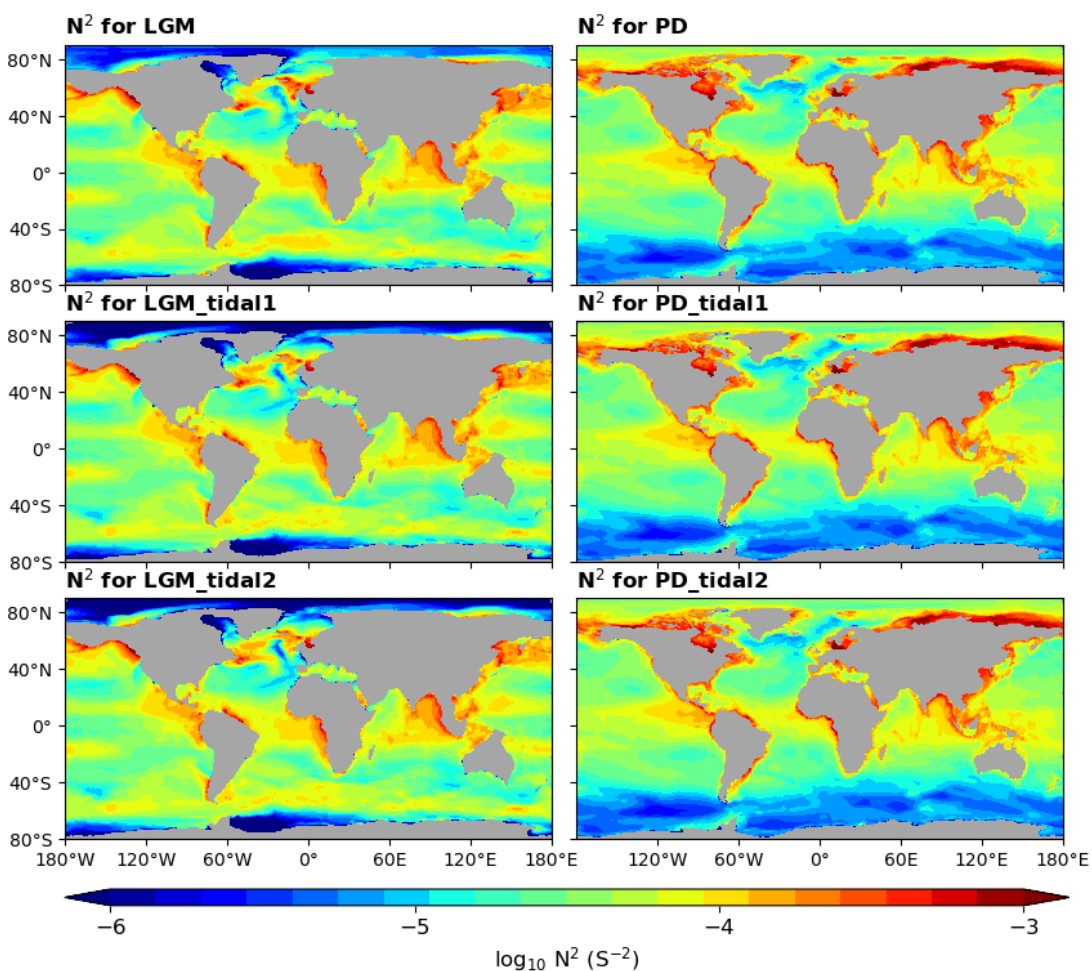

**Figure A1.** Changes in depth-averaged vertical $N^2$ Across Iterations for the LGM and PD Simulations.

**Code availability.** The FESOM 2.0 source code used for simulations in this study is available at https://github.com/FESOM/fesom2.git. The reanalysis data used for PD forcing in this study is the JRA-55 (the Japanese 55-year Reanalysis), which can be downloaded at https://jra.kishou.go.jp/JRA-55/index_en.html. The model output data used for LGM forcing in this study is available through Zhang et al. (2013, https://cp.copernicus.org/articles/9/2319/2013/).

**Author Contributions.** Y.C. and G.L. conceived and designed the study. Y.C. developed and performed the model simulations under the guidance of P.S., G.L., and X.C. And all authors contributed to the writing and revising of the manuscript.

**Competing interests.** The authors declare that they have no conflict of interest.

**Financial support.** X.C. is supported by the Natural Science Foundation of China under Grant 42394130 and the National Key R&D Program of China under Grant 2019YFA0607000. G.L. receives funding through "Ocean and Cryosphere under Climate Change" in the program "Changing Earth – Sustaining our Future" of the Helmholtz Society and through the Bundesministerium für Bildung und Forschung (Grant Nos. 01LP1917A and 01LP2004A).

**Acknowledgments.** We appreciate conversations with Dr. Dehai Song and Dr. Hu Yang during the course of this work. Furthermore, we extend our sincere thanks to the HPC department of the Alfred Wegener Institute (AWI) and the German Climate Computation Center (DKRZ) for their support. Special thanks to Handling Editor Marisa Montoya for her meticulous work, and to Guido Vettoretti and an anonymous reviewer for their constructive comments and suggestions.

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
