# Peer review of "Shoaled glacial AMOC despite vigorous tidal Dissipation: Vertical Stratification matters"

_Climate of the Past, 2024_

## Referee Comment (RC1)

Review of "Shoaled glacial AMOC despite vigorous tidal Dissipation: Vertical Stratification matters" by Chen et al.

This paper presents simulations of the stratification and overturning circulation in the LGM. The main point is that the AMOC is relatively shallow, despite the stronger tidal dissipation of the LGM, because the stratification also matters. A strong stratification prevents the AMOC from being deep. The strong tidal dissipation does create a stronger production of AABW relative to the present-day. The paper nicely combines simulations of tides, OGCM simulations of the general circulation, overturning, and stratification, and discussions of the literature on LGM conditions, e.g., Adkins et al. 2002. I think that the paper is a nice contribution and should eventually be published. I say this as someone who is a contributor to some of the tidal literature cited here, but who is not at all an expert on simulating the overturning circulation. I hope that some of the other reviewers will be familiar with the latter topic. Below I list a few  major points that I think should be improved, as well as some specific points.

**Major point 1.** First, there are a lot of feedbacks between stratification, ocean tides, and the OGCM, associated with equation (3). I think that both your procedure and the feedbacks could be better described. Equation (3) contains a factor of omega (tidal frequency). So, which frequency did you use? Probably the $M_2$ frequency, I'm guessing. Please state what you did. Similarly, the formula contains a factor of $N^2$, the very stratification that you are (presumably) getting from your OGCM simulations, which are affected by the tidal dissipation. And the tidal dissipation in turn is affected by your assumption of $N^2$. So there are lots of sensitivities here! Again, I think you should describe your procedure and what you did to address these sensitivities.

**Major point 2.** As far as I can tell, the atmospheric forcing employed here is not described at all. Surely, the wind and buoyancy forcing must matter? Otherwise the authors would be saying that one need change only the bathymetry and tidal forcing to get this dramatically different ocean, which would seem surprising, at least to me. At any rate, it would be very useful to describe the atmospheric forcing, which is always a critical factor in ocean modeling.

**Major point 3.** As noted below, there are some obvious places (in my opinion, at least) where references should be added. Also, I found some errors in the referencing (such as citing a different paper led by Harper Simmons than the one you intended) even with a fairly casual check of papers that I know very well. So this makes me wonder if the referencing might have some other similar errors. Please should check your references over more carefully to ensure that everything is accurately cited.

**Specific points**

Line 41—I believe that this is the first mention of the enhanced LGM tidal dissipation in the main body of the text. This would be a good place to mention that this finding of enhanced

LGM tidal dissipation has been found by many authors, beginning with the Egbert et al. 2004 paper that you cite elsewhere, and continuing in other papers (the Griffiths and Peltier papers, the Green 2010 paper, and others, many of which are already in your reference list). On line 199, you could state that your own results of enhanced LGM tidal dissipation are consistent with results from these earlier studies, and cite them again. How exactly you do it is up to you but you should cite this earlier work on this critical point.

Line 49—Arbic et al. 2004a reference should actually be Arbic et al. 2004b

Line 62—suggest removing "Actually" at the front of the sentence (unnecessary)

Line 72—I suggest describing zeta_EQ (the astronomical potential) first, and then describing alpha as a factor that corrects for the astronomical body tides (cite Hendershott 1972, which can also be cited for zeta_SAL)

Hendershott, M.C., 1972. The effects of solid earth deformation on global ocean tides. Geophys. J. R. Astron. Soc. 29, 389–402. https://doi.org/10.1111/j.1365-246X.1972.tb06167.x

Line 83—"The last 20 days are used for harmonic analysis". You also need to tell us how many days you ran for.

Line 84—"Is" should not be capitalized as it is in the middle of a sentence

Lines 85 and 86—please define what "node number" and "cell number" mean

Equation (5): the error is probably calculated over a tidal cycle; this should be stated. Also, why is there a "2" in the denominator; this is not usually present. Unless you are accounting for the factor of ½ in the time-average of a squared cosine function. The latter would mean that you are using harmonically analyzed amplitudes in equation (5) rather than instantaneous values; in which case you should say that. Bottom line, you could make this a bit more clear.

Table 1: the errors are reasonable. It would be good to compare them to errors attained by other forward tide models in the literature.

Line 117: there are two nice papers by Harper Simmons in 2004, and you are citing the wrong one. The paper that followed the parameterization of Jayne and St. Laurent 2001 is Simmons, Jayne, St. Laurent, Weaver 2004, Ocean Modelling, https://doi.org./10.1016/S1463-5**003(03)00011-8**. So you should replace the Simmons, Hallberg, Arbic 2004 citation in your references with the above reference.

Line 142—you apply five cycles. Five cycles of what? Please explain.

---

## Referee Comment (RC3)

Review of Chen et al. "Shoaled glacial AMOC despite vigorous tidal Dissipation: Vertical Stratification matters"

This is a nice result of one particular ocean model that shows only a slightly stronger LGM AMOC then present day, but more shoaled, using a stronger calculated LGM tidal mixing. Ferrari 2014, suggested that a shallower interface between North Atlantic Deep Water (NADW) and Antarctic Bottom Water (AABW), as observed for the LGM, reduced mixing between the two water masses, and in turn increased deep carbon. So this study can help test our understanding of biogeochemical cycles during deglaciation in further studies that come along. The impact of the strong LGM forcing on the mechanical forcing of the AMOC is novel. It would have been nice to see further theoretical analysis on how the mixing does not impact the AMOC in the upper ocean (e.g. meridional transport based upon zonal density gradients etc.). Also I find there isn't much effort put into describing the ocean model here and giving a more detailed description or illustration of how the ocean model actually performs against modern observations , in particular how the water masses compare and how the stratification compares in modern (see below).

L47-50: Add some more text here that there is more dissipation in the interior instead of the shelves at LGM. Removing the shelves reduces the damping of the tides and leads to increase in tidal amplitude.

L92: You might mention here that the use of ICE-6G instead of ICE-5G is suggested to reduce internal vertical mixing and would therefore suggest a further weakened AMOC (Wilmes et al 2021).

In Figure 3 (left column) I would like to see instead a vertical profile of horizontally averaged $N^2$ and the mean values. What mean values are used in the tide model $D\_IT$ (internal wave drag)? Are they taken from the PD and LGM simulations?

Section on Ocean Model: I would like to see more information about the ocean model described here. What is the resolution? In addition to "Figure S1 presents the horizontal resolution of the meshes for PD and LGM". please describe it here. How does it perform with respect to the present day? Isn't the AMOC a little weak compared with the RAPID array or other observations? How is the modern ocean forced? Does it use COREv2 AMIP type forcing? I want a better description of how the stratification compares with modern day observations. Maybe a T-S density showing different water masses in the ocean compared with modern observations (say ARGO). We don't get a good feel from this document on how the model actually performs against modern observations, which is the most important part of the paper.

Line 115: In ocean models "K_bg is employed to manage the effects of various background mixing mechanisms". However, the tidal mixing parameterization considers the locally dissipated energy over topography (⅓ of the total energy dissipation). The other ⅔ is dissipated in the far-field in which the background diffusivity is used to represent this. Is this correct interpretation? If so , wouldn't this tend to underestimate the effect of the increased tidal mixing.

Therefore, there is a constant internal energy dissipation due to internal wave breaking in the far field of something of the order of Integral ( Gamma^-1 * rho * N^2 kbg) dV. Do you know how large this value is?

Line 121: Kv_tidal is dependent on N^-2 and the internal tide dissipation energy, epsilon. Epsilon is increasing at LGM , but the stratification is also increasing . I would like to see a quantitative comparison of the results produced in Figure 2 (right column) due to each component , the internal tide energy dissipation and the stratification.

L125: One of the biggest problems I have is with these Jayne et al 2009 type parameterizations is the vertical decay function F using this universal e-folding factor of 500m. Would this not tend to overestimate the internal tide mixing energy in shallow seas? But if your hypothesis is true, then the LGM surface forcing overcomes these inadequacies in the parameterization.

L133: "which resonates with the North Atlantic basin". Do you mean that the predominant contributor , the M2 tide, has increased resonance in the North Atlantic at LGM. Isn't this due predominantly because of the removal of the shelves at LGM and the increases of tidal mixing in the deep North Atlantic Ocean at LGM?

L135-136: So the horizontal variations of D_IT are put into FESOM in a one off setting. I assume the variations in the Brunt Vaisala frequency are taken from an LGM simulation then used to calculate the LGM tides and then the ocean model is run with these. There would presumably be some feedback between the stratification and the tide model if this were done in a proper interactive way and that the results might be different if this were done. This should be mentioned here.

L142 expand on "we apply five cycles"

L147 Also the atmospheric forcing is held fixed. A reduced AMOC would have reduced heat transport to the North Atlantic which would favour sea ice growth to some degree, even though the atmosphere tends to compensate for the lack of ocean heat transport. This would presumably affect deep water formation and stratification. The same would happen around Antarctica. This limitation should be discussed or mentioned somewhere.
This atmospheric forcing aspect is, however, mentioned briefly in the conclusions.

L160: See comments in L121 above.

L170: In PD scenarios integrating the tidal mixing…

L172 Mention Table S1 here.

Discussions:

This is a nice result and this paper should be added to the literature on the subject. In particular as the discussions conclude, the study suggests that stronger stratification significantly reduces the impact of tidal dissipation. However, in the abyssal ocean with relatively weak stratification, the pronounced tidal dissipation during the LGM notably enhances the formation of AABW.

In paleoclimate settings, increased AABW production is often associated with a colder Antarctica and increased sea ice. From Table S1, some of the increase appears to be due to the atmospheric forcing.

Maybe add a comment that since the LGM atmospheric forcing is fixed, it separates out possible effects that would occur due interactions of Southern Ocean sea ice growth on AABW formation.

---

## Author Comment (AC1)

This paper presents simulations of the stratification and overturning circulation in the LGM. The main point is that the AMOC is relatively shallow, despite the stronger tidal dissipation of the LGM, because the stratification also matters. A strong stratification prevents the AMOC from being deep. The strong tidal dissipation does create a stronger production of AABW relative to the present-day. The paper nicely combines simulations of tides, OGCM simulations of the general circulation, overturning, and stratification, and discussions of the literature on LGM conditions, e.g., Adkins et al. 2002. I think that the paper is a nice contribution and should eventually be published. I say this as someone who is a contributor to some of the tidal literature cited here, but who is not at all an expert on simulating the overturning circulation. I hope that some of the other reviewers will be familiar with the latter topic. Below I list a few major points that I think should be improved, as well as some specific points.

We thank the reviewer for their positive and constructive feedback on our manuscript. Here are our point-by-point responses:

Major point 1. First, there are a lot of feedbacks between stratification, ocean tides, and the OGCM, associated with equation (3). I think that both your procedure and the feedbacks could be better described. Equation (3) contains a factor of omega (tidal frequency). So, which frequency did you use? Probably the $M_2$ frequency, I'm guessing. Please state what you did. Similarly, the formula contains a factor of $N^2$, the very stratification that you are (presumably) getting from your OGCM simulations, which are affected by the tidal dissipation. And the tidal dissipation in turn is affected by your assumption of $N^2$. So there are lots of sensitivities here! Again, I think you should describe your procedure and what you did to address these sensitivities.

**Authors' Response:** Thank you for your valuable suggestions. Indeed, our description of the forward tidal model was not sufficiently detailed. Yes, we used the frequency of the M2 tide equation (3).
Furthermore, as you rightly pointed out, there is indeed an interaction between $N^2$ in the tidal model and the OGCM: we input the $N^2$ obtained from the OGCM into the tidal model, and the resulting tidal dissipation in turn affects the $N^2$ in the OGCM.
To mitigate these effects and sensitivities, we used an iterative process as follows: Taking the LGM simulations as examples, we first input the $N^2$ obtained from the LGM case (no tidal mixing) into the tidal model, then input the resulting tidal dissipation back into the OGCM, obtaining the experimental result LGM_tidal1. Next, we input the $N^2$ from LGM_tidal1 back into the tidal model to obtain a new tidal dissipation, and run the OGCM again to obtain LGM_tidal2. The LGM_tidal shown in our manuscript is actually LGM_tidal2.
Following figure illustrates the changes in depth-averaged vertical $N^2$ during the iterations for both the PD and LGM. It can be seen that during the simulation of LGM, the change from LGM to LGM_tidal1 primarily involves a decrease in $N^2$ in the Arctic. From LGM_tidal1 to LGM_tidal2, there is almost no change. For the PD simulations, there were no significant changes in N2 throughout. Thus, we have nearly eliminated the mutual influences between $N^2$ in the tidal model and the OGCM through one iteration.

We acknowledge the need to enhance the description of the forward tidal model in the revised manuscript and will incorporate Figure 1 into the supplementary materials.

[Figure]

Figure 1. Changes in depth-averaged vertical N² Across Iterations for LGM and PD Simulations.

Major point 2. As far as I can tell, the atmospheric forcing employed here is not described at all. Surely, the wind and buoyancy forcing must matter? Otherwise the authors would be saying that one need change only the bathymetry and tidal forcing to get this dramatically different ocean, which would seem surprising, at least to me. At any rate, it would be very useful to describe the atmospheric forcing, which is always a critical factor in ocean modelling.

**Authors' Response:** Thank you for your feedback. As you rightly pointed out, atmospheric forcing is indeed crucial in ocean modeling. In our study, without considering tidal mixing, different atmospheric forcings have already resulted in distinct ocean between the present day (PD case) and the LGM (LGM case).
For the PD cases, we employed atmospheric forcing derived from the Reanalysis dataset (JRA55-do 1.4.0) spanning the period from 1958 to 2020. For the LGM cases, atmospheric results from a coupled climate model specifically tailored for the LGM (Zhang et al., 2013) were utilized to drive the ocean model.

Furthermore, the study by Knorr et al. (2021) highlights the variations in the temperature and salinity distributions of the ocean as simulated by different climate models for the LGM period. Not all models successfully capture the crucial characteristics of the glacial stratified ocean during the LGM. This underscores the critical importance of choosing an appropriate atmospheric forcing that can accurately reproduce the oceanic conditions during the LGM. Previous studies (Schmittner et al., 2015; Wilmes et al., 2019) have identified the phenomenon of enhanced tidal dissipation during the LGM contributing to a strong and deep AMOC, attributed to their failure to reproduce the high abyssal salinity and enhanced stratification in the LGM Atlantic.

In light of these findings, we will enhance our manuscript by including a more detailed description of the atmospheric forcing employed and further emphasize its crucial role in our ocean modeling.

Major point 3. As noted below, there are some obvious places (in my opinion, at least) where references should be added. Also, I found some errors in the referencing (such as citing a different paper led by Harper Simmons than the one you intended) even with a fairly casual check of papers that I know very well. So this makes me wonder if the referencing might have some other similar errors. Please should check your references over more carefully to ensure that everything is accurately cited.

**Authors' Response:** Thank you for your detailed feedback and for highlighting the issues with the references in my manuscript. I apologize for the oversight regarding the errors and omissions in the referencing. I have conducted a comprehensive review of all references cited in the manuscript. I will also add the necessary references in the sections you've pointed out as lacking.

**Specific points**

Line 41—I believe that this is the first mention of the enhanced LGM tidal dissipation in the main body of the text. This would be a good place to mention that this finding of enhanced LGM tidal dissipation has been found by many authors, beginning with the Egbert et al. 2004 paper that you cite elsewhere, and continuing in other papers (the Griffiths and Peltier papers, the Green 2010 paper, and others, many of which are already in your reference list). On line 199, you could state that your own results of enhanced LGM tidal dissipation are consistent with results from these earlier studies, and cite them again. How exactly you do it is up to you but you should cite this earlier work on this critical point.

**Authors' Response:** Thank you for your suggestion. I agree that it's important to reference the earlier work on enhanced LGM tidal dissipation when it's first mentioned and to cite these papers again when discussing our own results. I will make sure to include these references accordingly in the revised manuscript.

Line 49—Arbic et al. 2004a reference should actually be Arbic et al. 2004b.

**Authors' Response:** Thank you for pointing out the correction regarding the reference. I'll make sure to update it to "Arbic et al. 2004b" in the manuscript.

Line 62—suggest removing "Actually" at the front of the sentence (unnecessary)

**Authors' Response:** Thank you for the suggestion. I'll remove "Actually" from the beginning of the sentence as it's unnecessary, as you mentioned.

Line 72—I suggest describing zeta_EQ (the astronomical potential) first, and then describing alpha as a factor that corrects for the astronomical body tides (cite Hendershott 1972, which can also be cited for zeta_SAL)
Hendershott, M.C., 1972. The effects of solid earth deformation on global ocean tides. Geophys. J. R. Astron. Soc. 29, 389–402. https://doi.org/10.1111/j.1365- 246X.1972.tb06167.x

**Authors' Response:** Thank you for the suggestion. I will revise the description of zeta_EQ (the astronomical potential) first, followed by the description of alpha as a correction factor for astronomical body tides. Additionally, I'll cite Hendershott (1972) for both zeta_SAL and the correction factor alpha.

Line 83—"The last 20 days are used for harmonic analysis". You also need to tell us how many days you ran for.
**Authors' Response:** Thank you for your comment. Upon further inspection, the tidal model was indeed run for a total of 30 days, with the last 20 days being used for harmonic analysis. We will correct this error in the revised manuscript.

Line 84— "Is" should not be capitalized as it is in the middle of a sentence
**Authors' Response:** Thank you for catching that. I'll make sure to lowercase "Is" as it's in the middle of a sentence, as you pointed out.

Lines 85 and 86—please define what "node number" and "cell number" mean
**Authors' Response:** Thank you for the clarification. "Node" refers to the vertices of the unstructured triangular mesh, while "cell" refers to the triangles formed by connecting these nodes. I'll make sure to include this explanation in the manuscript.

Equation (5): the error is probably calculated over a tidal cycle; this should be stated. Also, why is there a "2" in the denominator; this is not usually present. Unless you are accounting for the factor of $1/2$ in the time-average of a squared cosine function. The latter would mean that you are using harmonically analyzed amplitudes in equation (5) rather than instantaneous values; in which case you should say that. Bottom line, you could make this a bit more clear.
**Authors' Response:** Thank you for your feedback. We acknowledge that the explanation in our manuscript was not sufficiently clear. Here, we are indeed using harmonically analyzed amplitudes (complex notation sinusoids) to evaluate the elevations, which accounts for the "2" in the denominator. We will clarify this methodology and provide a more detailed explanation in the revised manuscript.

Table 1: the errors are reasonable. It would be good to compare them to errors attained by other forward tide models in the literature.

**Authors' Response:** We agree that it would be beneficial to compare the errors attained by our model to those of other forward tide models in the literature. We will include following comparison table in the revised manuscript. Thanks!

Table. Comparison of M2 RMS error in Forward Tide Models

| Model | Deep Water (cm) | Shallow Water (cm) | Global (cm) |
|---|---|---|---|
| Our Tidal Model | 4.87 | 15.14 | 6.54 |
| Egbert et al. (2004) | < 5 | NA | NA |
| Arbic et al. (2004) | 7.26 | NA | NA |
| Griffiths and Peltier (2009) | NA | NA | 13.6 |
| Wilmes and Green (2014) | 3.86 | NA | 6.67 |
| Schindelegger et al. (2018) | 4.4 | 14.6 | NA |

*Note: Differences in the observed models selected across different studies may influence the results.

Line 117: there are two nice papers by Harper Simmons in 2004, and you are citing the wrong one. The paper that followed the parameterization of Jayne and St. Laurent 2001 is Simmons, Jayne, St. Laurent, Weaver 2004, Ocean Modelling, https://doi.org./10.1016/S1463-5003(03)00011-8. So you should replace the Simmons, Hallberg, Arbic 2004 citation in your references with the above reference.

**Authors' Response:** Thank you for bringing this to our attention. We will replace the citation for Simmons et al. (2004a) with the correct reference Simmons et al. (2004b), in the revised manuscript.

Line 142—you apply five cycles. Five cycles of what? Please explain.

**Authors' Response:** Thank you for your inquiry. For the PD cases, our surface (atmospheric) forcing is derived from the Reanalysis dataset (JRA55-do 1.4.0), covering the period from 1958 to 2020. We repeatedly drove each PD case with data from this time span five times to achieve simulation stability. We will include this description in the revised manuscript.

*Cited literature:*

Arbic, B. K., Garner, S. T., Hallberg, R. W., and Simmons, H. L.: The accuracy of surface elevations in forward global barotropic and baroclinic tide models, Deep Sea Research Part II: Topical Studies in Oceanography, 51, 3069-101, 10.1016/j.dsr2.2004.09.014, 2004.

Egbert, G. D., Ray, R. D., and Bills, B. G.: Numerical modeling of the global semidiurnal tide in the present day and in the last glacial maximum, Journal of Geophysical Research: Oceans, 109, 10.1029/2003jc001973, 2004.

Griffiths, S. D. and Peltier, W. R.: Modeling of Polar Ocean Tides at the Last Glacial Maximum: Amplification, Sensitivity, and Climatological Implications, Journal of Climate, 22, 2905-24, 10.1175/2008jcli2540.1, 2009.

Hendershott, M. C.: Effects of Solid Earth Deformation on Global Ocean Tides, Geophys J Roy Astr S, 29, 389-+, DOI 10.1111/j.1365-246X.1972.tb06167.x, 1972.

Knorr, G., Barker, S., Zhang, X., Lohmann, G., Gong, X., Gierz, P., Stepanek, C., and Stap, L. B.: A salty deep ocean as a prerequisite for glacial termination, Nat Geosci, 14, 930-+, 10.1038/s41561-021-00857-3, 2021.

Schindelegger, M., Green, J. A. M., Wilmes, S. B., and Haigh, I. D.: Can We Model the Effect of Observed Sea Level Rise on Tides?, J Geophys Res-Oceans, 123, 4593-609, 10.1029/2018jc013959, 2018.

Schmittner, A., Green, J. A. M., and Wilmes, S. B.: Glacial ocean overturning intensified by tidal mixing in a global circulation model, Geophys Res Lett, 42, 4014-22, 10.1002/2015gl063561, 2015.

Simmons, H. L., Hallberg, R. W., and Arbic, B. K.: Internal wave generation in a global baroclinic tide model, Deep-Sea Research Part Ii-Topical Studies in Oceanography, 51, 3043-68, 10.1016/j.dsr2.2004.09.015, 2004a.

Simmons, H. L., Jayne, S. R., St Laurent, L. C., and Weaver, A. J.: Tidally driven mixing in a numerical model of the ocean general circulation, Ocean Modelling, 6, 245-63, 10.1016/S1463-5003(03)00011-8, 2004b.

Wilmes, S. B. and Green, J. A. M.: The evolution of tides and tidal dissipation over the past 21,000 years, J Geophys Res-Oceans, 119, 4083-100, 10.1002/2013jc009605, 2014.

Wilmes, S. B., Schmittner, A., and Green, J. A. M.: Glacial Ice Sheet Extent Effects on Modeled Tidal Mixing and the Global Overturning Circulation, Paleoceanography and Paleoclimatology, 34, 1437-54, 10.1029/2019pa003644, 2019.

Zhang, X., Lohmann, G., Knorr, G., and Xu, X.: Different ocean states and transient characteristics in Last Glacial Maximum simulations and implications for deglaciation, Clim Past, 9, 2319-33, 10.5194/cp-9-2319-2013, 2013.

---

## Author Comment (AC3)

This is a nice result of one particular ocean model that shows only a slightly stronger LGM AMOC then present day, but more shoaled, using a stronger calculated LGM tidal mixing. Ferrari 2014, suggested that a shallower interface between North Atlantic Deep Water (NADW) and Antarctic Bottom Water (AABW), as observed for the LGM, reduced mixing between the two water masses, and in turn increased deep carbon. So this study can help test our understanding of biogeochemical cycles during deglaciation in further studies that come along. The impact of the strong LGM forcing on the mechanical forcing of the AMOC is novel. It would have been nice to see further theoretical analysis on how the mixing does not impact the AMOC in the upper ocean (e.g. meridional transport based upon zonal density gradients etc.). Also I find there isn't much effort put into describing the ocean model here and giving a more detailed description or illustration of how the ocean model actually performs against modern observations, in particular how the water masses compare and how the stratification compares in modern (see below).

We thank the reviewer for their positive and constructive feedback on our manuscript. Below are our detailed, point-by-point responses:

L47-50: Add some more text here that there is more dissipation in the interior instead of the shelves at LGM. Removing the shelves reduces the damping of the tides and leads to increase in tidal amplitude.

**Authors' Response:** Thank you for your valuable feedback. We will include a brief explanation on lines 47-50 to clarify that during the LGM, the reduction of the continental shelves led to decreased tidal damping and consequently enhanced tides.

L92: You might mention here that the use of ICE-6G instead of ICE-5G is suggested to reduce internal vertical mixing and would therefore suggest a further weakened AMOC (Wilmes et al 2021).

**Authors' Response:** Thank you for your valuable suggestions. Yes, the choice of LGM bathymetry databases can influence the tidal dissipation obtained in the tidal model. The tidal dissipation derived using ICE-6G is indeed weaker than that obtained using ICE-5G (Wilmes et al., 2019; Wilmes et al., 2021). We will add a corresponding explanation in the manuscript. Our choice to use ICE-5G is to demonstrate that even under conditions of strong tidal dissipation, the tides alone are insufficient to alter the shallower geometry of the AMOC during the LGM. We will include the corresponding explanations in the revised manuscript.

In Figure 3 (left column) I would like to see instead a vertical profile of horizontally averaged $N^2$ and the mean values. What mean values are used in the tide model $D_{IT}$ (internal wave drag)? Are they taken from the PD and LGM simulations?

**Authors' Response:** Thank you for your feedback. Yes, the $N^2$ values used in the tidal model are derived from the PD and LGM simulations. We have provided a global depth-averaged distribution of $N^2$ in response to your comment regarding L135-136.

Section on Ocean Model: I would like to see more information about the ocean model described here. What is the resolution? In addition to "Figure S1 presents the horizontal resolution of the meshes for PD and LGM". please describe it here. How does it perform with respect to the present day? Isn't the

AMOC a little weak compared with the RAPID array or other observations? How is the modern ocean forced? Does it use COREv2 AMIP type forcing? I want a better description of how the stratification compares with modern day observations. Maybe a T-S density showing different water masses in the ocean compared with modern observations (say ARGO). We don't get a good feel from this document on how the model actually performs against modern observations, which is the most important part of the paper.

**Authors' Response:** Thank you for your valuable comments. We will include Figure S1 in the main text and add relevant information regarding the resolution of the meshes. For the PD cases, we employed atmospheric forcing derived from the Reanalysis dataset (JRA55-do 1.4.0) spanning the period from 1958 to 2020.

Yes, the simulated AMOC is slightly weaker compared to observations. However, what is more crucial in our simulations is the ability to reproduce the geometry of the AMOC for both PD and LGM periods, as well as the ocean characteristics of these periods, and study the effects of tides on this basis. We have included a comparison of the temperature and salinity in the Atlantic Ocean from our PD case with the WOA (World Ocean Atlas) 2018 data, as shown in the Figure R1 below. The results indicate that our model effectively reproduces the temperature and salinity structures of the modern ocean. Based on this foundation, we accurately replicated the ocean characteristics during the LGM period as indicated by proxy data (Adkins et al., 2002; Knorr et al., 2021): strong vertical stratification caused by elevated salinity of the deep sea and the rapid temperature decrease in the ocean's upper layers. This was not achieved in previous studies(Schmittner et al., 2015; Wilmes et al., 2019), which is why they concluded that tides would significantly enhance the AMOC during the LGM.

[Figure]

**Figure R1.** Comparison of salinity and temperature between WOA 18 data and PD simulation in the Atlantic Ocean.

Line 115: In ocean models "$k_{bg}$ is employed to manage the effects of various background mixing mechanisms". However, the tidal mixing parameterization considers the locally dissipated energy over topography (1⁄3 of the total energy dissipation). The other 2⁄3 is dissipated in the far-field in which the background diffusivity is used to represent this. Is this correct interpretation? If so, wouldn't this tend to underestimate the effect of the increased tidal mixing.

Therefore, there is a constant internal energy dissipation due to internal wave breaking in the far field of something of the order of Integral (Gamma^-1 * rho * $N^2$ $k_{bg}$) dV. Do you know how large this value is?

**Authors' Response:** Thank you for your suggestions. The tidal mixing parameterization indeed only considers the locally dissipated energy. Therefore, we calculated the far-field dissipation due to $k_{bg}$ and the local tidal dissipation for each simulation using Osborn (1980) formula:

$$P = \int \rho \epsilon \ \mathrm{dV} = \frac{1}{\Gamma} \int \rho k \, N^2 \ \mathrm{dV}$$

The results are shown in the table R1 below. Additionally, we conducted a new experiment, LGM_tidal_3, where we increased $k_{bg}$ from $1*10^{-5}$ $m^2s^{-1}$ to $3*10^{-5}$ $m^2s^{-1}$ for comparison. The results in the table indicate that the LGM_tidal experiment indeed underestimated the energy provided by tides, while the total energy in LGM_tidal_3 reached 3.92 TW.

We compared the AMOC geometry in LGM_tidal and LGM_tidal_3, as shown in the figure below. The geometry of the AMOC in LGM_tidal_3 does not show significant changes and remains relatively shallow, further supporting the conclusions of our study. The only notable change is in the AMOC strength, which increased from 13.3 Sv to 15.4 Sv. We will add this discussion to the revised manuscript. We believe that these additions emphasize not only the necessity of employing tidal mixing parameterization but also the importance of appropriately adjusting the background diffusivity $k_{bg}$.

**Table R1.** Summary of energy consumption due to diapycnal mixing

| Simulation | $k_{bg}$ ($1*10^{-5}$ $m^2s^{-1}$) | $k_{bg}$ contribution (TW) | Tidal contribution (TW) | Total (TW) |
|---|---|---|---|---|
| PD | 1 | 0.79 | 0 | 0.79 |
| PD_tidal | 1 | 0.78 | 0.38 | 1.16 |
| LGM | 1 | 1.05 | 0 | 1.05 |
| LGM_tidal | 1 | 1.02 | 1.03 | 2.05 |
| LGM_tidal_3 | 3 | 2.86 | 1.06 | 3.92 |

[Figure]

**Figure R2.** AMOC streamfunctions (Sv) between LGM_tidal and LGM_tidal_3.

Line 121: $K_{v\_tidal}$ is dependent on $N^2$ and the internal tide dissipation energy, epsilon. Epsilon is increasing at LGM, but the stratification is also increasing. I would like to see a quantitative comparison of the results produced in Figure 2 (right column) due to each component, the internal tide energy dissipation and the stratification.

**Authors' Response:** The change in tidal dissipation has specific figures, increasing from 1.31 TW during the PD to 3.41 TW during the LGM. For $N^2$, we will give the percentage increase both globally and in the Atlantic Ocean, comparing the changes between these two parameters.

It is worth noting that there is significant spatial distribution variability between them. Therefore, we further plotted the vertical distribution of the rate of tidal dissipation. Figure R3 presents the zonally averaged vertical distributions of the rate of tidal dissipation in the Atlantic Ocean. However, due to the spatial variability of dissipation (as shown in Figure 1 of the manuscript) and changes in water depth, the dissipation rate in Figure R3 shows considerable variation and does not clearly reflect its decrease from the seafloor upwards. Therefore, in Figure R4, we provide the rate of tidal dissipation (left), the squared buoyancy frequency (middle), and tidal diffusivity (right) along the 27°W section.

[Figure]

**Figure R3.** Zonally averaged vertical distributions of the rate of tidal dissipation for PD and LGM in the Atlantic Ocean.

[Figure]

**Figure R4.** The rate of tidal dissipation (left), the squared buoyancy frequency (middle), and tidal diffusivity (right) along 27°W section.

L125: One of the biggest problems I have is with these Jayne et al 2009 type parameterizations is the vertical decay function F using this universal e-folding factor of 500m. Would this not tend to overestimate the internal tide mixing energy in shallow seas? But if your hypothesis is true, then the LGM surface forcing overcomes these inadequacies in the parameterization.

**Authors' Response:** Regarding the issues with the tidal mixing parameterization, I would like to address this based on my understanding. In the barotropic tidal model, tidal energy is dissipated in two forms: the bottom friction term (equation 2 in the manuscript) and the internal-wave drag term (equation 3 in the manuscript). The dissipation of $D_{BL}$ caused by the former primarily occurs in shallow seas, while $D_{IT}$ caused by the latter mainly occurs in deep seas. The global distribution is shown in Figure R5. The proportion of $D_{IT}$ used in tidal mixing parameterization in regions shallower than 500 meters is very small, as indicated in Table R1.

**Table R1.** Global and Sub-500m Distribution of $D_{IT}$ and $D_{BL}$ during PD and LGM.

| Time | $D_{IT}$ | $D_{IT}$ (<500 m) | $D_{BL}$ | $D_{BL}$ (<500 m) |
|------|----------|-------------------|----------|-------------------|
| PD   | 1.31     | 0.21              | 1.69     | 1.65              |
| LGM  | 3.41     | 0.36              | 1.17     | 0.99              |

[Figure]

**Figure R5.** Global distributions of bottom friction dissipation ($D_{BL}$) and internal-tide dissipation ($D_{IT}$) during PD and LGM.

L133: "which resonates with the North Atlantic basin". Do you mean that the predominant contributor, the M2 tide, has increased resonance in the North Atlantic at LGM. Isn't this due predominantly because of the removal of the shelves at LGM and the increases of tidal mixing in the deep North Atlantic Ocean at LGM?

**Authors' Response:** Thank you for your feedback. Actually, 12.66 hours is one of the natural resonant periods of the present-day Atlantic Ocean (Muller, 2008), which is very close to the 12.42 hours period of the M2 tide. In this resonant situation, the removal of the shelves during the LGM led to decreased damping, resulting in a significant increase in the M2 tide. I will include this explanation in the revised manuscript.

L135-136: So the horizontal variations of $D_{IT}$ are put into FESOM in a one off setting. I assume the variations in the Brunt Vaisala frequency are taken from an LGM simulation then used to calculate the LGM tides and then the ocean model is run with these. There would presumably be some feedback between the stratification and the tide model if this were done in a proper interactive way and that the results might be different if this were done. This should be mentioned here.

**Authors' Response:** Thank you for your comment. In fact, we used an iterative approach to address the interaction between $D_{IT}$ and ocean simulations for the PD_tidal and LGM_tidal experiments. The iterative process is as follows: Taking the LGM simulations as examples, we first input the $N^2$ obtained from the LGM case (no tidal mixing) into the tidal model, then input the resulting tidal dissipation back into the OGCM, obtaining the experimental result LGM_tidal1. Next, we input the $N^2$ from LGM_tidal1 back into the tidal model to obtain a new tidal dissipation, and run the OGCM again to obtain LGM_tidal2. The LGM_tidal shown in our manuscript is actually LGM_tidal2.

Figure R6 illustrates the changes in depth-averaged vertical $N^2$ during the iterations for both the PD and LGM. It can be seen that during the simulation of LGM, the change from LGM to LGM_tidal1 primarily involves a decrease in $N^2$ in the Arctic. From LGM_tidal1 to LGM_tidal2, there is almost no change. For the PD simulations, there were no significant changes in N2 throughout. Thus, we have nearly eliminated the mutual influences between $N^2$ in the tidal model and the OGCM through one iteration.

[Figure]

**Figure R6.** Changes in depth-averaged vertical N² Across Iterations for LGM and PD Simulations.

L142 expand on "we apply five cycles"

**Authors' Response:** Thank you for your inquiry. For the PD cases, our surface (atmospheric) forcing is derived from the Reanalysis dataset (JRA55-do 1.4.0), covering the period from 1958 to 2020. We repeatedly drove each PD case with data from this time span five times to achieve simulation stability. We will include this description in the revised manuscript.

L147 Also the atmospheric forcing is held fixed. A reduced AMOC would have reduced heat transport to the North Atlantic which would favour sea ice growth to some degree, even though the atmosphere tends to compensate for the lack of ocean heat transport. This would presumably affect deep water formation and stratification. The same would happen around Antarctica. This limitation should be discussed or mentioned somewhere.

This atmospheric forcing aspect is, however, mentioned briefly in the conclusions.

**Authors' Response:** Thank you for pointing out this important aspect. You are correct that holding the atmospheric forcing fixed could limit the representation of feedbacks between the AMOC, heat transport, and sea ice growth. We acknowledge this limitation and will include it in the revised manuscript.

L160: See comments in L121 above.

**Authors' Response:** As mentioned in the response to L21, we will provide a quantitative comparison.

L170: In PD scenarios integrating the tidal mixing...

**Authors' Response:** Apologies for the ambiguity. I will revise this sentence to: In the PD_tidal experiment, incorporating tidal mixing parameterization does not alter the geometry of the AMOC.

L172 Mention Table S1 here.

Thank you for your suggestion. We will mention Table S1.

Discussions:

This is a nice result and this paper should be added to the literature on the subject. In particular as the discussions conclude, the study suggests that stronger stratification significantly reduces the impact of tidal dissipation. However, in the abyssal ocean with relatively weak stratification, the pronounced tidal dissipation during the LGM notably enhances the formation of AABW.

In paleoclimate settings, increased AABW production is often associated with a colder Antarctica and increased sea ice. From Table S1, some of the increase appears to be due to the atmospheric forcing. Maybe add a comment that since the LGM atmospheric forcing is fixed, it separates out possible effects that would occur due interactions of Southern Ocean sea ice growth on AABW formation.

Thank you for the positive evaluation. Yes, one of the important reasons for the increase in AABW during the LGM is indeed the colder background climate, which has been discussed in many papers. Therefore, we did not elaborate on this point in detail. The Figure R7 below shows the sea ice extent and MLD in the Southern Ocean for several cases in this study. It is evident that the background climate or the LGM atmospheric forcing predominantly results in a larger sea ice extent, which further facilitated the formation of AABW. At the same time, the strong tides during the LGM significantly further promoted the formation of AABW under this background.

We will add a statement in the manuscript to address the limitations of our ocean-only simulations, specifically noting that the interactions of Southern Ocean sea ice growth and AABW formation might be overlooked due to the fixed LGM atmospheric forcing.

[Figure]

**Figure R7.** Southern Ocean Mixed Layer Depth (MLD) and 50% sea ice concentration contour line (black) for different experiments.

*Cited literature:*

Adkins, J. F., McIntyre, K., and Schrag, D. P.: The salinity, temperature, and delta O-18 of the glacial deep ocean, Science, 298, 1769-73, DOI 10.1126/science.1076252, 2002.

Knorr, G., Barker, S., Zhang, X., Lohmann, G., Gong, X., Gierz, P., Stepanek, C., and Stap, L. B.: A salty deep ocean as a prerequisite for glacial termination, Nat Geosci, 14, 930-+, 10.1038/s41561-021-00857-3, 2021.

Muller, M.: Synthesis of forced oscillations, Part I: Tidal dynamics and the influence of the loading and self-attraction effect, Ocean Modelling, 20, 207-22, 10.1016/j.ocemod.2007.09.001, 2008.

Osborn, T. R.: Estimates of the Local-Rate of Vertical Diffusion from Dissipation Measurements, J Phys Oceanogr, 10, 83-9, Doi 10.1175/1520-0485(1980)010<0083:Eotlro>2.0.Co;2, 1980.

Schmittner, A., Green, J. A. M., and Wilmes, S. B.: Glacial ocean overturning intensified by tidal mixing in a global circulation model, Geophys Res Lett, 42, 4014-22, 10.1002/2015gl063561, 2015.

Wilmes, S. B., Green, J. A. M., and Schmittner, A.: Enhanced vertical mixing in the glacial ocean inferred from sedimentary carbon isotopes, Commun Earth Environ, 2, ARTN 166 10.1038/s43247-021-00239-y, 2021.

Wilmes, S. B., Schmittner, A., and Green, J. A. M.: Glacial Ice Sheet Extent Effects on Modeled Tidal Mixing and the Global Overturning Circulation, Paleoceanography and Paleoclimatology, 34, 1437-54, 10.1029/2019pa003644, 2019.